# Deconvoluting complex structural histories archived in brittle fault zones

G. Viola[1,2,†], T. Scheiber[1,†], O. Fredin[1,3], H. Zwingmann[4], A. Margreth[1] & J. Knies[1,5]

Brittle deformation can saturate the Earth's crust with faults and fractures in an apparently chaotic fashion. The details of brittle deformational histories and implications on, for example, seismotectonics and landscape, can thus be difficult to untangle. Fortunately, brittle faults archive subtle details of the stress and physical/chemical conditions at the time of initial strain localization and eventual subsequent slip(s). Hence, reading those archives offers the possibility to deconvolute protracted brittle deformation. Here we report K-Ar isotopic dating of synkinematic/authigenic illite coupled with structural analysis to illustrate an innovative approach to the high-resolution deconvolution of brittle faulting and fluid-driven alteration of a reactivated fault in western Norway. Permian extension preceded coaxial reactivation in the Jurassic and Early Cretaceous fluid-related alteration with pervasive clay authigenesis. This approach represents important progress towards time-constrained structural models, where illite characterization and K-Ar analysis are a fundamental tool to date faulting and alteration in crystalline rocks.

[1] Geological Survey of Norway, Trondheim 7491, Norway. [2] Department of Geology and Mineral Resources, Norwegian University of Science and Technology, Trondheim 7491, Norway. [3] Department of Geography, Norwegian University of Science and Technology, Trondheim 7491, Norway. [4] Department of Geology and Mineralogy, Kyoto University, Kitashirakawa Oiwake-cho, Kyoto 606-8502, Japan. [5] CAGE—Centre for Arctic Gas Hydrate, Environment and Climate, Department of Geology, The Arctic University of Norway, Tromsø 9037, Norway. † Present address: Department of Biological, Geological and Environmental Sciences, BiGeA, University of Bologna, 40126 Bologna, Italy (G.V.); Faculty of Engineering and Science, Sogn og Fjordane University College, Sogndal 6851, Norway (T.S.). Correspondence and requests for materials should be addressed to G.V. (email: giulio.viola3@unibo.it).

Faults are more than just the visible evidence of momentous brittle deformation. They are unique archives of upper-crustal tectonic histories and, importantly, of the prevailing physical and chemical conditions at the time of deformation. This is because, once formed, faults may be extremely sensitive to variations of the stress field and environmental conditions and are readily prone to slip and reactivation in a variety of settings, also in regions affected by only weak, far-field stresses. However, unravelling the details of the complete deformation history archived by a fault can be challenging. The reason is the commonly chaotic framework of brittle deformation, which may reflect multiple factors including the high number of accumulated faults and fractures of different age that can literally saturate the Earth's upper crust, the generally hard to decipher nature of poorly textured brittle fault rocks, the significant fluid-rock interaction that may take place within the faults' dilatant cores and, above all, the ease of reactivation of brittle structures[1]. In fact, particularly in the case of geologically old faults, continued reactivation may lead to the almost complete obliteration of the early strain increments, which hampers efforts aiming at an efficient use of a fault's archive.

Despite these difficulties, it is now generally acknowledged that careful, multiscalar structural analysis can produce reliable geometric and kinematic/dynamic models of brittle deformation histories spanning over several hundreds of millions years, even in geologically complex and multiply-deformed terrains[2–6]. Moreover, some authors have successfully constrained the first-order time dimension of brittle deformation by $^{40}Ar/^{39}Ar$ and K-Ar dating of synkinematic mica/illite from brittle-ductile and brittle fault rocks[7–24]. Uncertainties, however, still remain as to the way to interpret these results, mostly reflecting the incomplete understanding of the mechanisms of clay crystallization and growth associated with the physical and chemical evolution of fault rocks. In addition, most dating studies tend to overlook the structural complexity of the dated faults to focus instead on advancing the geochronological component of the research and, as a result, it is generally difficult to link the obtained ages to the subtle details of the structural evolution recorded by the fault.

Aiming at bridging this gap, we propose a multidisciplinary approach, which, by combining K-Ar dating of carefully selected brittle fault rocks with the comprehensive structural characterization of the fault architecture, allows the construction of time-constrained brittle evolutionary models. We present an hitherto unreported brittle extensional fault from western Norway, the Goddo Fault, whose reconstructed deformation history is now shown to span a 200 Myr time interval, from initial nucleation to reactivation and to final alteration due to later and structurally compartmentalized fluid flow. Multiple illite size fractions from fault samples representative of very different (yet coexisting) structural domains were dated by K-Ar analysis and characterized by petrography, scanning and transmission electron microscopy (SEM, TEM) and X-ray diffraction (XRD). We use the distinct shape of the inclined 'grain-size versus age' spectra[25–27] of our study to gain further insight in the processes of clay authigenesis during brittle deformation and to support a conceptual model of general validity for the K-Ar dating of illite from brittle fault rocks. In summary, we demonstrate an innovative methodological approach to untangle deformational histories, which would otherwise remain hidden within the commonly complex, if not chaotic, architecture of brittle faults and fault rocks. This is in turn crucial to regional tectonic reconstructions and/or better understanding of the processes that steer fault localization and growth as a function of the delicate and time-dependent interplay between hardening and weakening mechanisms.

## Results

**Geological setting and sample description.** The study area is located onshore southwestern Norway in the Scandinavian

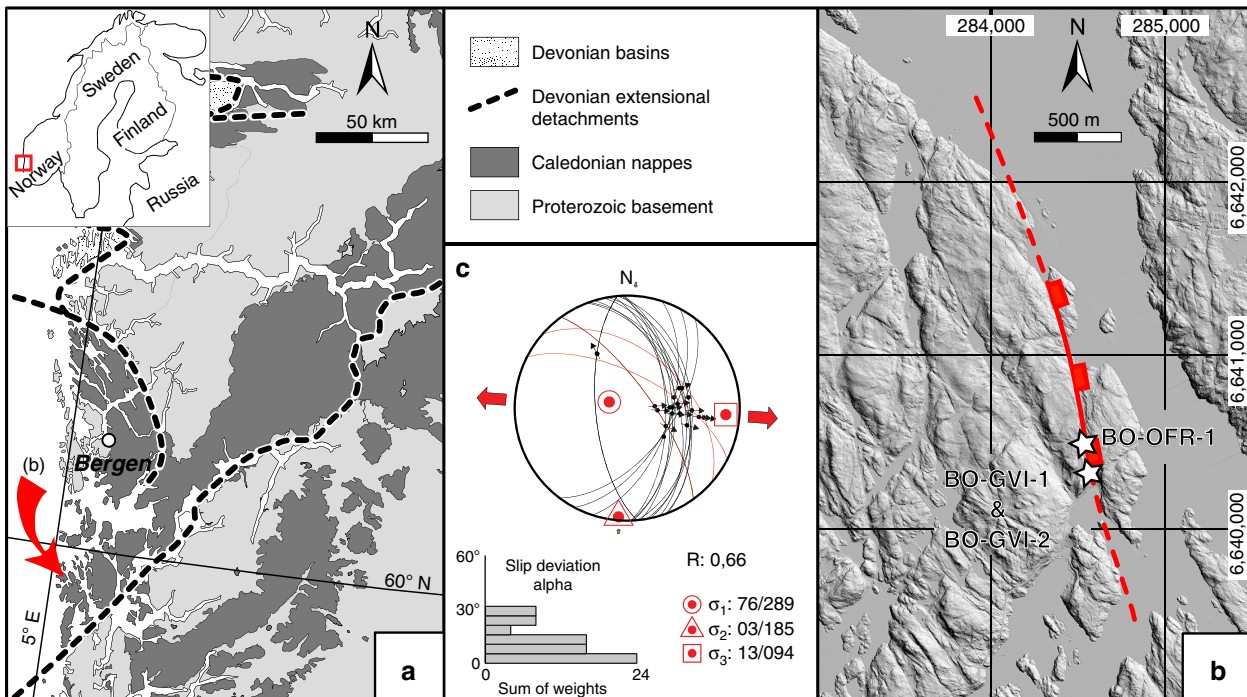

**Figure 1 | Goddo Fault location and inversion of its fault-slip data.** (**a**) Location map of the study area in western Norway. Arrow indicates Goddo Island. (**b**) Hill-shaded LiDAR digital elevation model (DEM) of the southeasternmost Goddo Island illuminated from the northeast. Red line: trace of the Goddo Fault. (**c**) Stress tensor derived from the inversion of the quartz (± hematite) striated Goddo Fault principal slip surface (black great circles) and older, reactivated NW-SE-striking epidote-coated strike-slip faults (red great circles; equal area, lower hemisphere projection; inversion run with Win-Tensor[60]). E-W extension is indicated.

Caledonides (Fig. 1a), which formed in the Silurian-Early Devonian during overall NW-SE convergence between Baltica and Laurentia[28,29]. During the Early Devonian, crustal-scale extensional shear zones accommodated the NW-SE collapse of the by then over-thickened orogenic pile, dramatically attenuating the crust and causing uplift of the orogenic roots. In response to progressive uplift and denudation, the extensional deformation style evolved progressively from purely ductile to brittle[30]. By the Permo-Triassic, faulting was mostly brittle and associated with doleritic dyke intrusion along N-S to NNW-SSE structural trends[5,6,31–34]. Brittle extension and reactivation of pre-existing structures continued through the Jurassic to the Early Cretaceous, reflecting crustal thinning and final rifting in the North Sea[32,35,36]. The relative tectonic quiescence associated with thermal subsidence and major sediment loading, which characterized the Early Cretaceous in the Norwegian North Sea, was followed by inversion, localized uplift and structural reactivation induced by far-field compression generally ascribed to the Alpine orogeny[37] and, more importantly, the Cretaceous evolution of the Mid-Norwegian margin[38–40].

Apatite fission track and (U-Th)/He data from the study area and its immediate surroundings constrain overall Permian to Late Cretaceous cooling, with the majority of the dates clustering between the Late Triassic and the Early Cretaceous[41–43]. These are generally interpreted as reflecting a first phase of rapid exhumation in the Triassic followed by a second in the Jurassic, likely assisted and facilitated by significant faulting. The region, moreover, is not believed to have been exposed to burial with temperatures in excess of c. 60 °C since the Early Jurassic[42,43].

The Goddo Fault is exposed along a ~500 m long N-S road section on the eastern Goddo Island (Fig. 1b). It deforms the 466 ± 3 Ma, texturally homogeneous, medium-grained and locally porphyritic Rolvsnes granodiorite[44,45]. The fault is characterized by a large, smooth and gently undulating, NNW-SSE striking and moderately ~E-dipping principal slip surface (PSS; Figs 1c and 2a,b). The PSS is relatively continuous and can be followed over large tracts of the section. It is characterized by a striated fault mirror and second-order Riedel shears decorated primarily by fine-grained quartz (Fig. 2c), with subordinate hematite, epidote and pyrite. Fault-slip data from dip-slip quartz and quartz/hematite striations and reactivated epidote striations constrain a ~E-W reduced extensional stress tensor (Fig. 1c). The PSS post-dates and discordantly cuts through steep NW-SE trending striated fractures, presumably legacy of an earlier transpressional deformation episode of Late Ordovician age[45]. Some of these older fractures presently bear two different striation sets (Fig. 2d): the steeper striations are in quartz (red line in Fig. 2d) and are interpreted as documenting the reactivation during the Goddo Fault overall E-W extension of these earlier, subvertical strike-slip and epidote-decorated fractures that contain a set of older, more gently-plunging striations (blue lines in Fig. 2d).

The PSS is generally overlain by a c. 0.5–1 m thick layer of immature, indurated cataclasite (Fig. 2a), which is transitional to the heavily fractured and in places strongly altered hanging wall. At one locality, however, the preserved fault core is more complex and is formed by two distinct and spatially separated structural domains, each with a different fault rock type (Fig. 2b). Immediately above the PSS there is an up to ~0.3 m thick

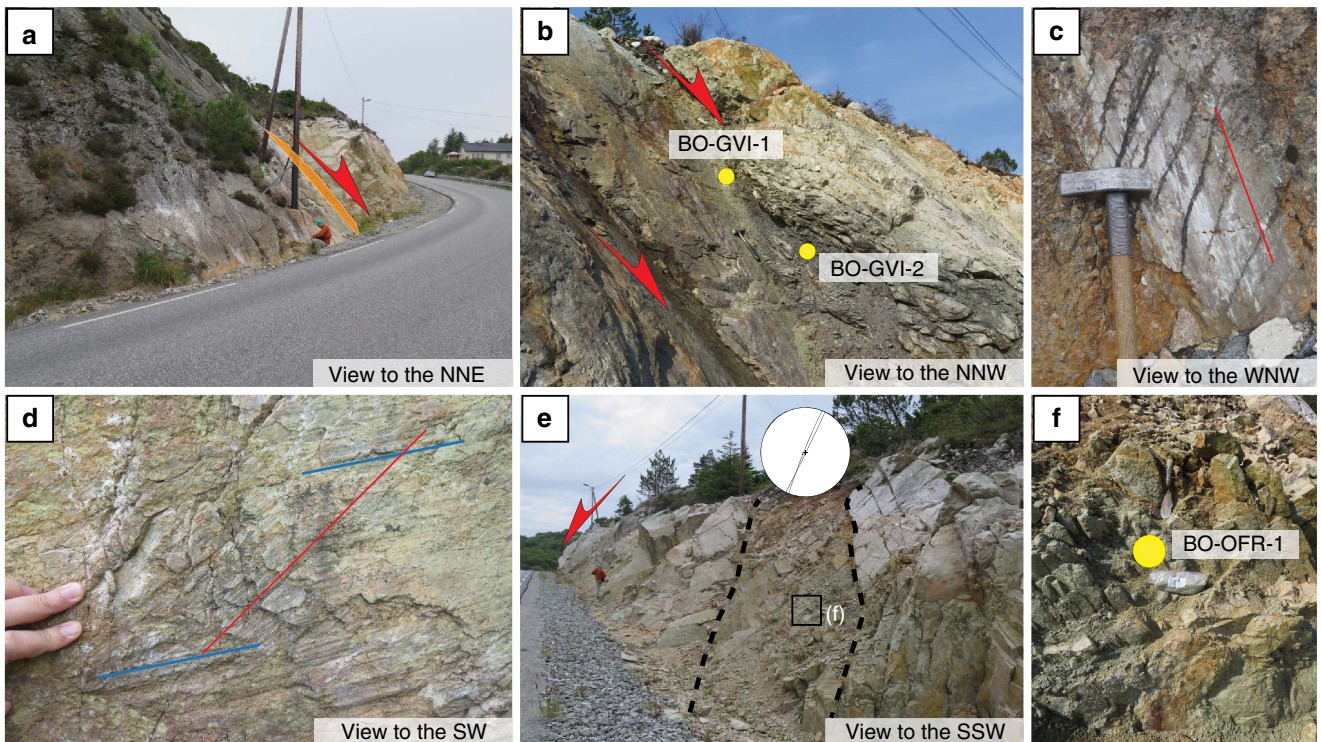

**Figure 2 | Goddo Fault in the field.** (**a**) Sharp principal slip surface (PSS) overlain by 0.5–1 m of cataclasite (orange stripe). T. Scheiber for scale. (**b**) PSS overlain by greenish clay-rich gouge (sample BO-GVI-1) and phyllonite (sample BO-GVI-2), indicating top-to-the-E extensional kinematics. Hammer for scale. (**c**) Example of quartz (± hematite) striated PSS. (**d**) Example of pre-extension steeply dipping strike-slip fault plane, with steeper quartz striations (red) overprinting earlier and flatter epidote striae (blue). (**e**) Black dashed lines define the NNE-SSW fracture corridor dissecting the Goddo Fault PSS. (**f**) Sampling site for sample BO-OFR-1 consisting of strongly altered granodiorite. Note the subrounded shape of the granodiorite altered knobs, reminiscent of core stone formation and incipient saprolitization. Note spatula for scale (c. 200 mm long) in the upper part of the outcrop.

greenish gouge layer. The gouge is parallel to the PSS, is fine-grained, well sorted and rich in clay minerals. No obvious kinematic indicators were observed in it. Sample BO-GVI-1 is from this plastic gouge (Fig. 2b). Above the green gouge, the fault core is characterized by a transitional phyllonite with a distinctive

grey-reddish color, passing upward to the fractured and locally intensely bleached hanging wall. The phyllonitic level is very gently undulated, and locally some closer and slightly asymmetric folds indicate an east vergence. The phyllonitic S/C' fabrics indicate invariably a top-to-the-E sense of shear, corresponding, at the present orientation, to E-W extension. Quartz and calcite veinlets parallel to the foliation are common. Sample BO-GVI-2 is from this fault rock.

Farther north, the PSS is cut discordantly by a distinctive, highly fractured zone (Fig. 2e). Its bounding fractures strike NNE-SSW and define a $\sim 2\,m$ wide corridor of severely altered granodiorite (Fig. 2f). Some striated quartz-coated slip planes occur within the fractured and altered domain, constraining bulk WNW-ESE to NW-SE oblique extension. No fault rock is present, possibly indicative of the low amount of strain accommodated along these fractures. The granodiorite still preserves its primary magmatic texture, but pervasive feldspar alteration has caused disaggregation of the rock, with significant growth of authigenic clay and a residual sandy quartz matrix. Decimetric subrounded core stones are present. Sample BO-OFR-1 is from the authigenic clay-rich altered part of the outcrop (Fig. 2f).

**K-Ar dating results.** The analytical procedures to separate, characterize and date illite are described by Zwingmann, et al.[24] Five clay size fractions were separated and dated for each sample, ranging from $<0.1\,\mu m$ to $6-10\,\mu m$ for the finest and coarsest, respectively, with $<0.4$, $<2$ and $2-6\,\mu m$ as intermediate sizes (Table 1). Ages are reported to the timescale of Gradstein, et al.[46] For sample BO-GVI-1, ages range between $200.2 \pm 4.1$ Ma for the

**Table 1 | K-Ar data.**

| Sample ID | Grain size fraction (μm) | K (%) | Rad. $^{40}Ar$ (mol g$^{-1}$) | Rad. $^{40}Ar$ (%) | Age (Ma) | Error (Ma) |
|---|---|---|---|---|---|---|
| BO-GVI-1 | <0.1 | 4.95 | 1.8174E-09 | 83.33 | 200.2 | 4.1 |
| BO-GVI-1 | <0.4 | 4.83 | 1.9499E-09 | 87.47 | 218.9 | 4.4 |
| BO-GVI-1 | <2 | 4.42 | 1.9738E-09 | 88.47 | 240.7 | 5 |
| BO-GVI-1 | 2-6 | 2.98 | 1.5182E-09 | 95.66 | 272.1 | 5.5 |
| BO-GVI-1 | 6-10 | 2.96 | 1.5727E-09 | 96.97 | 282.9 | 5.7 |
| BO-GVI-2 | <0.1 | 4.06 | 2.0028E-09 | 87.78 | 264.1 | 5.4 |
| BO-GVI-2 | <0.4 | 4.09 | 2.0002E-09 | 88.18 | 262 | 5.3 |
| BO-GVI-2 | <2 | 4.85 | 2.7829E-09 | 91.73 | 303.8 | 6.3 |
| BO-GVI-2 | 2-6 | 5.74 | 3.8401E-11 | 97.62 | 349.6 | 7.1 |
| BO-GVI-2 | 6-10 | 5.95 | 4.0410E-09 | 97.98 | 354.4 | 7.1 |
| BO-OFR-1 | <0.1 | 0.25 | 5.5968E-11 | 24.2 | 125.2 | 4.2 |
| BO-OFR-1 | <0.4 | 0.21 | 4.5538E-11 | 18.31 | 121.4 | 5.3 |
| BO-OFR-1 | <2 | 0.27 | 8.1700E-11 | 27.42 | 167.7 | 6.5 |
| BO-OFR-1 | 2-6 | 0.4 | 2.0225E-10 | 67.79 | 272.1 | 5.8 |
| BO-OFR-1 | 6-10 | 0.42 | 2.2651E-10 | 74.38 | 287.5 | 6.2 |

**Table 2 | K-Ar age standards and airshot data.**

| Standard | K (%) | Rad. $^{40}Ar$ (mol g$^{-1}$) | Rad. $^{40}Ar$ (%) | Age (Ma) | Error (Ma) | % Difference from recommended reference age |
|---|---|---|---|---|---|---|
| HD-B1–122 | 7.96 | 3.3607E-10 | 92.17 | 24.19 | 0.38 | − 0.08 |
| HD-B1–124 | 7.96 | 3.3776E-10 | 92.58 | 24.31 | 0.37 | + 0.41 |
| LP6–136 | 8.37 | 1.9254E-9 | 97.24 | 127.98 | 1.98 | + 0.06 |
| LP6–138 | 8.37 | 1.9237E-09 | 96.70 | 127.87 | 1.90 | − 0.02 |

| Airshot ID | $^{40}Ar/^{36}Ar$ | $+/-$ |
|---|---|---|
| AS118-AirS-1 | 295.17 | 0.56 |
| AS120-AirS-1 | 295.55 | 0.29 |

**Table 3 | XRD data.**

| Sample ID (μm) | Quartz | Kaolin | Illite/Mica 2M$_1$ | Illite/Mica 1M | Dioctahedral smectite | Interstratified illite/smectite | Albite/ Anorthite | Anatase |
|---|---|---|---|---|---|---|---|---|
| BO-GVI-1<0.1 | | 1 | 20 | | | 79 | | |
| BO-GVI-1<0.4 | | 5 | 21 | | 6 | 68 | | |
| BO-GVI-1<2 | 3 | 16 | 28 | | 24 | 29 | | <1 |
| BO-GVI-1 2-6 | 34 | 8 | 14 | | 13 | 30 | 1 | |
| BO-GVI-1 6-10 | 35 | 8 | 17 | | 12 | 27 | 1 | |
| BO-GVI-2 <0.1 | <1 | 12 | 14 | 4 | 20 | 49 | 1 | |
| BO-GVI-2 2-6 | 2 | 9 | 32 | 2 | 16 | 35 | 4 | |
| BO-GVI-2 6-10 | 2 | 9 | 33 | 4 | 17 | 31 | 4 | |
| BO-OFR-1<0.1 | | 7 | | | | 93 | <1 | |
| BO-OFR-1<0.4 | | 7 | | | | 93 | <1 | |
| BO-OFR-1<2 | | 16 | | | | 82 | 2 | |
| BO-OFR-1 2-6 | <1 | 22 | 3 | | | 71 | 3 | <1 |
| BO-OFR-1 6-10 | <1 | 25 | 2 | | | 68 | 4 | 1 |

XRD errors are estimated at 3%.

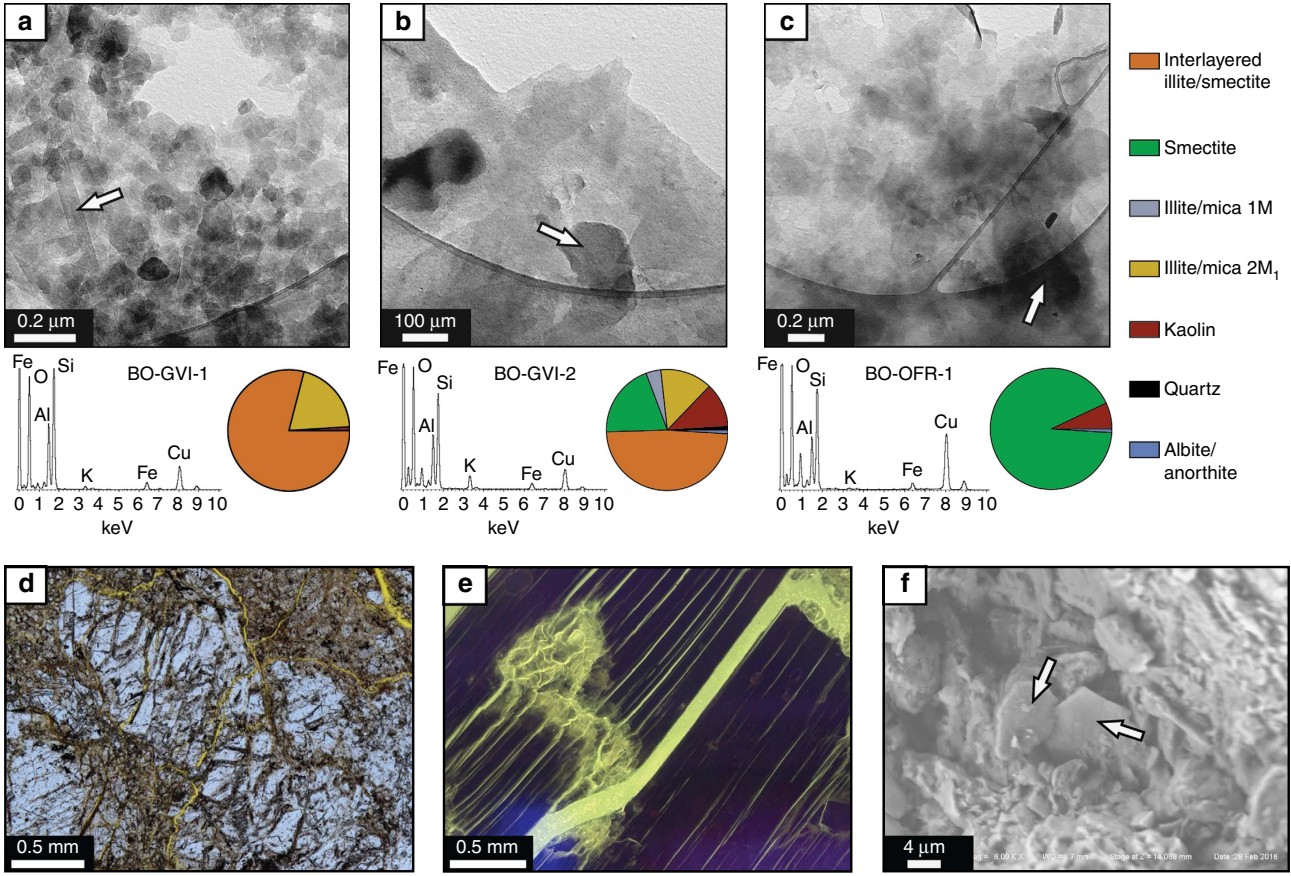

**Figure 3 | Characterization of the dated samples.** (**a**), (**b**) and (**c**) TEM images of authigenic/synkinematic illite with associated EDS spectra and XRD data for the <0.1 µm fraction of the three dated samples. Illite occurs in all samples with predominantly platy habit. (**d**) Cataclastic feldspar clast from the gouge sample BO-GVI-1. Note the high fracture density within the clast and the fine-grained groundmass. (**e**) Fluorescence image from sample BO-OFR-1 of biotite grain with dilation and fluid infiltration concentrated along the cleavage planes and a penetrating fracture network. (**f**) SEM image of platy illite within the finest fraction of sample BO-OFR-1.

finest and 282.9 ± 5.7 Ma for the coarsest fraction, corresponding to the Early Jurassic and the Permian (Cisuralian), respectively. The ages of sample BO-GVI-2 vary between 264.1 ± 5.4 Ma (Permian (Guadalupian)-finest) and 354.4 ± 7.1 Ma (Carboniferous (Mississipian)—coarsest). Sample BO-OFR-1 yielded ages between 125.2 ± 4.2 Ma (Early Cretaceous) for the finest <0.1 µm fraction and 287.5 ± 6.2 Ma (Permian (Cisuralian) for the coarsest fraction. K-Ar standard and airshot data for this study are summarized in Table 2. K content varies between c. 3 and 6% for the fault rock samples, but is remarkably lower for the altered granodiorite, ranging between 0.21 and 0.42% (Table 1).

Standard petrography, XRD analysis (Table 3), SEM and TEM imaging (Fig. 3a–c) allowed detailed characterization of the separated and dated fractions. From a compositional viewpoint, the clay-rich gouge sample BO-GVI-1 is heterogeneous, although its composition becomes mineralogically more homogenous in the finest fraction, in which only interstratified illite/smectite (79%) and illite/mica (20%) are present (Table 3 and Fig. 3a). Feldspar therein is pervasively fractured and crystals are fragmented by cataclastic flow (Fig. 3d). Phyllonite sample BO-GVI-2 contains the same mineral phases in all the three analysed fractions, though in different proportions. The finest <0.1 µm fraction is formed by interstratified illite/smectite (49%), dioctahedral smectite (20%), illite/mica 2M₁ (14%), kaolin (12%) and traces of the lower temperature 1M illite polytype (4%), albite/anorthite (1%) and quartz (<1%; Fig. 3b and Table 3). Illite/mica $2M_1$ content decreases from the coarsest

(33%) to the finest fraction (14%). The finest fraction of the strongly altered BO-OFR-1 granodiorite sample contains almost exclusively smectite (93%) with only minor kaolin (7%; Fig. 3c), indicative of advanced alteration of feldspar. Primary biotite is altered and shows evidence of bulk dilation (Fig. 3e), which provides positive feedback to fluid flow and further alteration. Although illite was not detected by XRD in the finest fraction of BO-OFR-1 (Table 3), TEM and SEM imaging revealed fine-grained illitic crystallites (for example, Fig. 3c,f). In all fifteen dated fractions, illite and interstratified illite/smectite are the only K-bearing phases. SEM and TEM analyses (Fig. 3) confirm that most illite crystallites in the finest fractions are characterized by well-preserved prismatic platy habit, suggesting in situ neocrystallization[47,48].

## Discussion

The dynamic nature of faults, which is the expression of rapidly evolving and often transient processes such as friction, fluid flow and rheological changes within generally dilatant rock volumes, is such that their architecture and composition evolve progressively through time and in space[9]. This leads to the at times apparently chaotic juxtaposition of structural domains whose composition, texture, physical properties and also isotopic signature are the legacy of different stages of the fault's history. The Goddo Fault is a representative example of this complexity, as it has a variable fault core architecture, with distinct structural domains outcropping side by side, each characterized by a different fault

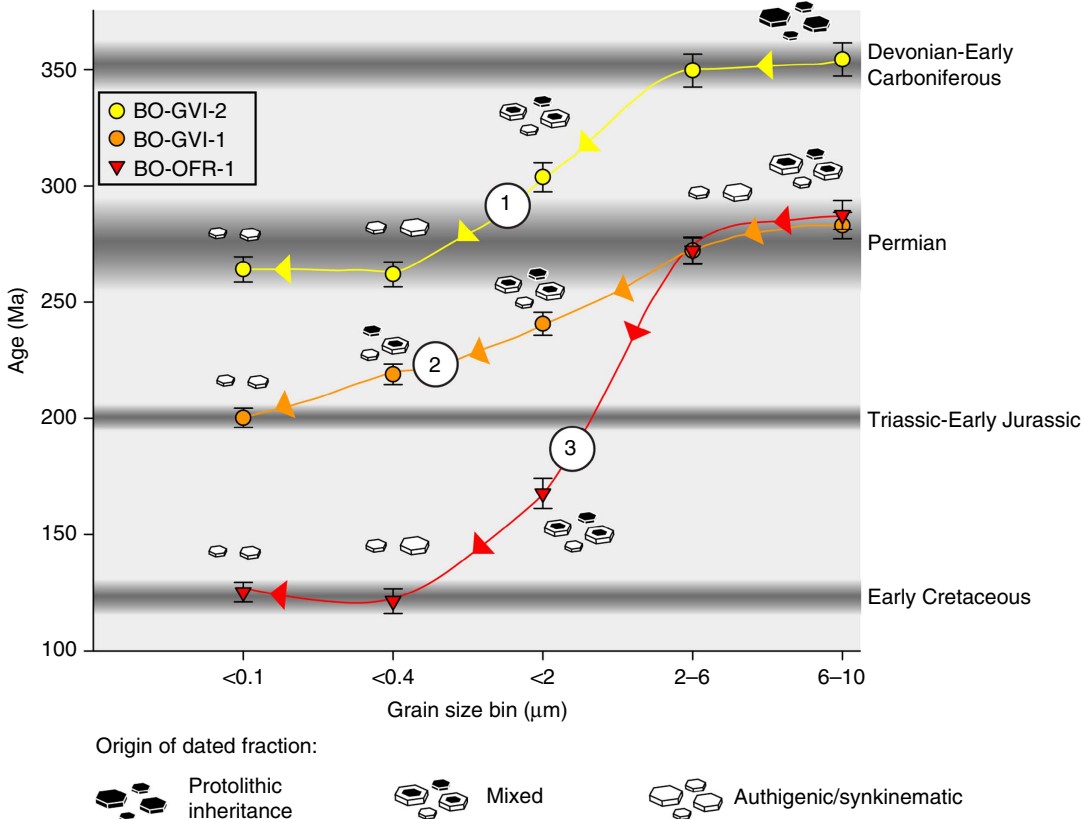

**Figure 4 | K-Ar age versus grain size spectra.** Dark grey horizontal bars: broadly defined time periods of significant thermal anomalies, faulting or rock alteration. The Permian event was characterized by major thermal anomalies, doleritic dyke intrusion and, later, increased cooling rates associated with significant, c. E-W extensional brittle-ductile faulting. During the Triassic-Early Jurassic continued dyke intrusion accompanied renewed E-W crustal stretching accommodated by entirely brittle structures. During the Early Cretaceous, localized transtension permitted significant fluid ingress into actively dilating rock volumes leading to deep rock alteration and clay authigenesis within the Goddo Fault core. The Age Attractor model[10,26] concludes that the K-Ar age of the finest illite fraction is the age of the last faulting episode recorded by the illite data. Note the plateaus at both ends for mixing curves 1 and 3 and for the coarsest fractions for curve 2, showing that illite authigenesis caused newly formed crystallites to grow to more than just the <0.1 μm fraction. All errors in the figure are shown as ± 2 sigma.

rock assemblage. Obviously, any structural interpretation leading to inferences on the regional tectonic history or the strength evolution of the fault and the implications thereof on, for example, seismogenesis, has to be validated against the temporal dimension of faulting. Neglecting the age of the individual faulting events responsible for the final fault architecture can result in erroneous conceptual models. Direct age constraints from the juxtaposed structural domains in a brittle fault are, in summary, crucial for the meaningful deconvolution of the complex history of initial strain localization and subsequent reactivation.

K-Ar illite dating of fault rocks is in general a robust method as it avoids $^{39}$Ar recoil and non-standardized encapsulation for $^{40}$Ar/$^{39}$Ar dating[19,49]. There is growing consensus that the K-Ar ages of the <0.1 μm (or finer) illite fraction date the latest episode of deformation recorded by the fault rock through growth of synkinematic and authigenic illite during faulting[8,13,20,21,23,24,26,50]. The age of the coarsest fractions is instead generally interpreted as representing a protolithic component, that is, an inherited input to the fault rock derived from the host rock[13,20]. This may be mica or illite derived from the previous cooling history of the rock or from an earlier faulting or fluid flow/alteration episode. Zwingmann et al.[27] reported the case of the Cretaceous Deokpori thrust in South Korea where synkinematic authigenesis is apparently recorded by both the <0.1 and <0.4 μm grain size fractions,

whereas the protolithic inherited signature was found in the <2, 2–6 and 6–10 μm fractions. Based on the geochronological results, they argued indirectly for complex clay growth patterns and selective preservation during deformation. Torgersen et al.[10,26] further developed the conceptual understanding of K-Ar age versus grain size relationships by proposing the Age Attractor model, according to which the age of the last recorded increment of deformation acts as an attractor towards which a mixing line converges from the oldest protolithic age. The resulting inclined age spectrum[25] thus reflects the mixing of inherited components with authigenic mineral phases and the slope of the spectrum is a function of the age difference between the two age end members. As a consequence, ages of intermediate grain size fractions in inclined spectra can be generally considered mixed ages reflecting authigenic growth of younger illitic rims on older protolithic nuclei as well as physical mixing of the two end members and are mostly devoid of geological meaning.

Our results (Fig. 4) confirm the validity of the Age Attractor model, but, in addition, improve the understanding of the underpinning theoretical background, thus justifying the Age Attractor model for general utilization. Our inclined spectra deviate from a classic linear age versus grain size mixing line, and instead define two plateaus where the ages of the two finest and/or coarsest fractions are identical within error. Curve 1 in Fig. 4, for example, suggests that faulting in the Permian (c. 264 Ma) acted as an attractor for a protolithic Carboniferous

component. The latter is documented not only by the coarsest 6–10 μm fraction, but also by the 2–6 μm fraction, showing that faulting in the Permian caused illite authigenesis and reworking of an isotopic system that had been well set in the Carboniferous, even for large grain sizes. The geological significance of the Carboniferous ages remains unclear. We exclude, however, that they reflect post-emplacement cooling of the Rolvsnes granodiorite because muscovite and biotite $^{40}$Ar/$^{39}$Ar ages constrain cooling of the pluton to the Middle Ordovician at about 460 Ma[45]. No deformation episodes have been reported in the literature for that time period, with existing $^{40}$Ar/$^{39}$Ar data from large extensional detachments constraining extensional deformation in the region to the Early and Middle Devonian[51,52] (c. 400 Ma). Eide et al.[53] on the other hand, report an episode of rapid cooling genetically connected to unroofing in Late Devonian-Carboniferous time (360–340 Ma) constrained by diffusion modelling of alkali feldspar from the Nordfjord-Sogn Detachment Zone, north of the study area. In the Bergen area, Larsen et al.[5] constrained hydrothermal alteration during the 371–363 Ma interval by Rb-Sr two-point isochrone dating of alkali-feldspar genetically connected with infiltration of hot fluids during Late Devonian-Carboniferous fracturing and faulting. Those results are broadly consistent with the c. 350 Ma age of the coarsest illite of sample BO-GVI-2 (Fig. 4, curve 1), possibly establishing a genetic link between the coarse fractions dated in this study and that episode of regional thermal underplating and tectonics.

Permian faulting must have been associated with stable and sufficient heat, fluid flow and strain to allow illite authigenesis and growth not only up to the <0.4 μm grain size within the phyllonitic fault rock of the Goddo Fault (sample BO-GVI-2), but also up to the coarsest dated fraction in the Goddo Fault clay-rich gouge (BO-GVI-1; curve 2) and the altered granodiorite (BO-OFR-1; curve 3). Whereas the protolithic end member ages of curves 2 and 3 are identical within error and broadly overlap with the Permian faulting age of curve 1, mixing lines 2 and 3 diverge significantly within the range of the <2 μm size fraction (Fig. 4). Curve 2 is attracted by a major faulting event of Late Triassic-Early Jurassic age as constrained by the date derived from the <0.1 μm fraction, while mixing line 3 is attracted by a process of younger Early Cretaceous illite authigenesis. Moreover, the latter forms a well-defined plateau where the age of the <0.1 and <0.4 μm fractions are identical within error (c. 121 and 125 Ma), indicating indeed significant illite authigenesis in the Early Cretaceous.

In summary, curves 1 and 3 allow the interpretation of the obtained age versus grain size spectra in terms of a step-wise process in which an initial faulting episode of Permian age caused widespread authigenesis from the finest to the coarsest dated fractions. The only exception to this pattern is the <2 μm fraction, which possibly reflects a higher amount of physical mixing between protolithic and authigenic/synkinematic crystallites. The coarsest fractions acted in turn as the common protolithic source whose isotopic and physical reworking generated the inclined spectra of curves 2 and 3 during the Triassic-Early Jurassic and Early Cretaceous structural events.

We exclude significant effects of thermally activated radiogenic $^{40}$Ar volume diffusion on the age of the finest grain size fractions and so interpret the inclined spectra as true mixing lines reflecting mainly varying amounts of authigenesis. Our conclusion relies on the results of numerical modelling that investigated potential partial $^{40}$Ar resetting within clay-size crystallites by, for example, Zwingmann, et al.[24] and Torgersen, et al.[10] Although performed with different numerical codes and different diffusion models, their results consistently show that radiogenic $^{40}$Ar diffusion curves obtained for relatively coarse mineral grains (such as

biotite or muscovite) during episodes of long and gradual cooling do not apply to clays formed within short-lived faults characterized by transient high temperatures and important fluid-rock interaction. Moreover, although radiogenic $^{40}$Ar volume diffusion does indeed occur, it has been concluded that it does not cause more than a 10% resetting of an K-Ar age during heating pulses up to 230–240 °C and lasting up to 10 Ma[10], even for very fine-grained illites (<0.1 μm). As mentioned earlier, the lack of ductile deformation in the study area and the available thermochronological data[41–43] suggest that the Goddo Fault was never exposed to a temperature higher than c. 220–250 °C.

The XRD data also support our interpretation of the results and can be used to propose a retrograde evolution of the Goddo Fault from the Devonian-Carboniferous. The high temperature $2M_1$ illite polytype[54] is found in both the phyllonite of sample BO-GVI-2 and the gouge of sample BO-GVI-1, consistent with formation of both fault rock types under similar thermal conditions within the brittle deformational regime. The $2M_1$ illite polytype, however, currently coexists with dioctahedral smectite and interstratified illite/smectite, which form and are commonly stable at lower temperature than illite. A likely explanation is the progressive smectitization of the primary illite (still present in its $2M_1$ polytype and only traces of the metastable lower-temperature variety 1M) during the polyphase and retrograde (that is, down to temperatures well below 200 °C) structural reactivation of the fault. Consistent with the retrograde smectitization of illite is also the simpler mineralogical composition of sample BO-OFR-1 (the youngest), which is essentially only composed of dioctahedral smectite and kaolin, indicating formation under the lowest recorded temperature associated with the Early Cretaceous event. This interpretation is, moreover, consistent with existing thermochronological studies[42,43] that suggest rapid Permo-Triassic cooling rates followed by slower exhumation and residence at temperatures lower than c. 60 °C from the Early Jurassic.

When the new geochronological results are coupled with the fault structural characteristics, the data suggest that the Goddo Fault documents cumulative strain localization during a retrograde deformation history that led to neocrystallization of at least two generations of synkinematic illite. Preservation in the dated fault rocks of pristine protolithic and/or only partially reset illite (protolithic and mixed ages, respectively) argues in favour of heterogeneous strain localization within the fault core such that multiscalar domains derived from the earlier deformation history are still selectively preserved because they escaped subsequent structural obliteration or pervasive overprint (Fig. 5). Thus, the fact that the coarsest fractions of sample BO-GVI-2 were not reset and maintain their Carboniferous isotopic signature (curve 1, Fig. 4) is strong evidence of heterogeneous strain localization associated with highly compartmentalized fluid flow controlled by the compositional and textural anisotropy of the fault core (Fig. 5b). Indeed, the physical separation in the field of the two dated fault rocks (with the clay-rich gouge immediately above the PSS and below the phyllonite) shows that brittle faulting caused disruption of the previously formed phyllonitic core by cataclastic flow (Fig. 5b). The PSS is possibly the result of a seismic sequence[55,56] that accommodated renewed extensional deformation after the main phase of gouge development, as also indicated by local, small E-dipping striated planes cutting the cataclastic layer overlying the PSS.

The Permian and Early Jurassic faulting events are both well constrained and dated. The identical top-to-the-E extensional kinematics, constrained by field observations for both fault rocks, show that the Goddo Fault nucleated, widened and eventually slipped along the PSS during a long-lived episode of c. E-W coaxial extension. This fits the known regional tectonic

evolution[32]. The Carboniferous-Permian was indeed associated with major lithospheric stretching offshore western Norway and north of the Tornquist fault system, eventually leading to the Oslo Rift with extensive volcanism and later uplift and emplacement of batholiths[57,58]. The presented new Permian ages agree with the Permian age of coast-parallel doleritic dykes in southwestern Norway[33], confirming major rifting. This deformational style is known to have protracted into the Triassic and the Early Jurassic[36]. In addition, both the Permian and Early Jurassic faulting phases identified in the Goddo Fault are identical within error to the K-Ar illite ages by Torgersen et al.[26] for a set of top-to-the-E extensional brittle faults immediately west of the Oslo Rift in the Kongsberg area, and thus stress the regional importance of this tectonic episode.

The Early Cretaceous age and the mineralogical characteristics of the altered granodiorite suggest fluid ingress and significant fluid-rock interaction along and within a NNE-SSW fracture zone that cuts through the Goddo Fault PSS and fault core. Despite the record of localized slip and dilation therein, no fault rock formed and clay authigenesis is entirely due to in-situ break-down of feldspar. It is noteworthy that the North Sea right offshore the study area is commonly described as having experienced relative tectonic quiescence during the Early Cretaceous, due to the progressive northward migration of the rifting activity towards the Mid-Norwegian margin during the Jurassic. The Early Cretaceous is instead reported as the time of important extension and associated exhumation related to hyperextension for the Mid-Norwegian segment of the margin[38,39]. Crustal stretching rotated progressively from ∼E-W in the North Sea to ∼NW-SE in the Mid-Norwegian margin. The Early Cretaceous age of the altered sample cutting discordantly through the N-S trending Goddo Fault thus tracks fluid ingress and chemical alteration along fractures that are discordant to the main North Sea structural grain, but are geometrically and kinematically compatible with the far-field stress effects of Cretaceous rifting in the mid-Norwegian margin north of the North Sea (Fig. 5c).

In conclusion, our results demonstrate the suitability of the proposed approach to constrain complex faulting histories in old crystalline basement blocks, where the lack of well-dated lithological markers generally prevents the reconstructions of geological histories.

The 'age versus grain size' inclined spectra reported here are unusual because of the presence of (sub)plateaus at the lower and higher end of the mixing lines defining the spectra, where the two finest (or coarsest) fractions have analytically identical ages. This is an ideal situation towards a flat age spectrum, which is a rare scenario for illite from brittle fault rocks, and argues for very significant synkinematic authigenesis during faulting. In multi-grain size fraction analyses of brittle fault rocks, the age of the finest is thus confirmed to be the closest to the age of the last recorded faulting episode.

The successful dating of the alteration of the Goddo Fault during the Early Cretaceous by means of K-Ar analysis of illite formed during fluid flow-related feldspar break-down[59] represents a useful step forward towards the full characterization of complex deformational processes in a time-constrained conceptual scheme.

The full structural characterization of the Goddo Fault has permitted to reconstruct a long-lasting history of strain localization during coaxial extension from the Permian to the Early Jurassic and later Early Cretaceous alteration during fluid ingress associated with renewed deformation. During the Late Triassic-Early Jurassic the faulting style evolved through a retrograde textural transition from phyllonitic fault rocks to clay-rich gouge. The associated kinematics indicate E-W extension for the Permian and Jurassic events, which confirms results from the offshore domain[32] and adds strong direct quantitative constraints to otherwise more qualitative studies in western Norway[5,6].

Our study demonstrates convincingly that K-Ar dating of authigenic illite can be successfully used to date both brittle faulting and alteration processes far back in geological time in crystalline basement rocks. It is important to stress, however, that our interpretation, which indeed discloses many details of the complex history of one single fault, is only possible because of the multi-grain size analysis carried out on multiple samples of the preserved structural domains within the Goddo Fault complex architecture. In addition, a comprehensive structural study of the dated faults remains necessary. The proposed combined approach is of general validity and due to the widespread occurrence of brittle features can lead to significant advances in the study of brittle structural geological processes and regional tectonic evolutions in any geological context.

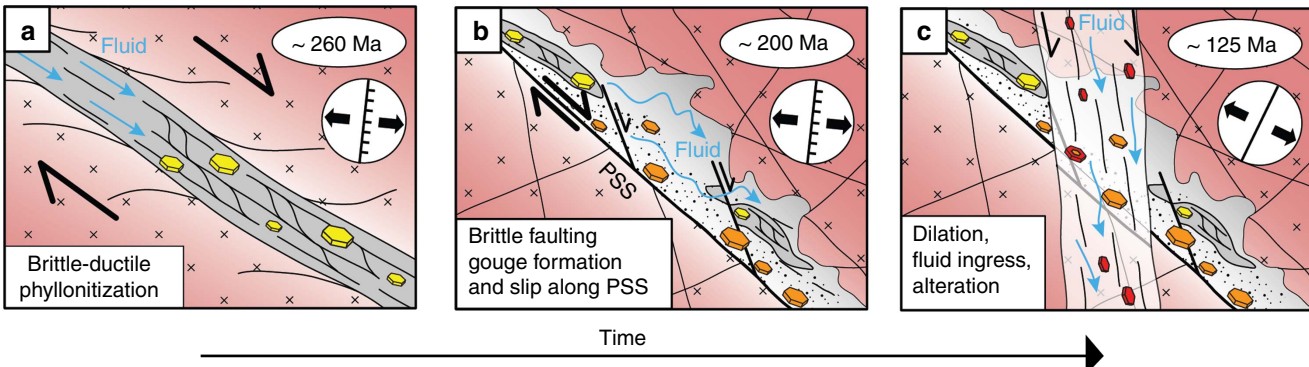

**Figure 5 | Conceptual evolutionary scheme of the Goddo Fault.** Each of the represented stages was characterized by illite authigenesis. Selective preservation within the fault core of structural domains representative of all deformation stages made it possible to separate, characterize and date three different illite generations (illite colour: yellow: sample BO-GVI-2; orange: sample BO-GVI-1; red: sample BO-OFR-1) and to deconvolute the complex and long-lasting evolution of the Goddo Fault archive. (**a**) Permian faulting led to a phyllonitic, N-S trending normal fault that accommodated overall E-W to ESE-WNW extension. (**b**) Coaxial extension continued through the Triassic and into the Jurassic. By then the faulting style was entirely brittle with gouge formation and localized slip along a striated PSS. (**c**) The far-field stress effects related to NW-SE extension along the Mid-Norwegian margin caused localized fracturing, dilation, significant fluid ingress and fluid-rock interaction, leading to Early Cretaceous authigenesis of a third illite generation.

**Data availability**. The authors declare that the data supporting the findings of this study are available within the paper.

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

## Acknowledgements

This study was conducted under the auspices of the BASE project, a research initiative at the NGU funded by Maersk Oil, Lundin Petroleum, Det Norske Oljeselskap, Wintershall and the NGU. J.K. is supported by the Research Council of Norway (NRC grant 223259). Sample separation and analyses were carried out at the CSIRO laboratories in Perth, Australia. Andrew Todd and Mark Raven, CSIRO, are thanked for technical assistance during the course of the study. Stress inversion results were obtained using Win-Tensor, a software developed by Dr Damien Delvaux, Royal Museum for Central Africa, Tervuren, Belgium. Espen Torgersen is thanked for his critical reading of an earlier version of the text. Constructive reviews by Kyle Min, Telemaco Tesei and Anna Ksienzyk improved significantly the paper.

## Author contributions

G.V., O.F. and J.K. conceived the study. H.Z. dated illite, acquired XRD diffractograms and produced SEM and TEM images. T.S. carried out field work with G.V. and O.F. and together with G.V. completed the structural analysis. A.M. prepared thin sections and measured SEM and elemental composition in some samples. G.V. wrote the paper with input from all authors. T.S. and G.V. prepared the figures.

## Additional information

**Competing financial interests:** The authors declare no competing financial interests.

