## [Peer Review File · Nature Communications]

Reviewers' comments:

Reviewer #1 (Remarks to the Author):

Review of the manuscript "Deconvoluting complex structural histories archived in brittle fault zones", by Viola et al.

The pioneering work of van der Pluijm et al. (2001, Nature), who found a general trend in Ar/Ar age - 2M_d/1M (detrital illite %) plot, provided a fundamental basis for constraining the timing of latest illite authigenesis. Their primary argument is that each fault gouge sample is composed of two endmembers: (1) a protolithic component whose K/Ar system is not affected by authigenesis, and (2) an authigenic component formed during late stage fault activity. Based on the positive, linear relationship in age - detrital illite %, they concluded that the y-intercept represents the timing of illite authigenesis, presumably during the latest fault activity. In later works, the Michigan group (Solum et al., 2005, J Struct Geol) developed more convincing evidence showing two linear trends with differential slopes, but identical y-intercept in the age - detrital illite % plot, reinforcing the previous conclusion that the finest size fraction yields the timing of latest authigenesis. Similar or slightly modified approaches have been applied for fault gouge samples.

This manuscript provides detailed structural data and K/Ar ages to constrain timing of faulting and fluid flow that occurred at a relatively shallow crustal section. Based on the general increase of K/Ar ages with size fractions, it was concluded that the finest size fractions yield the most reliable K/Ar ages corresponding to the timing of the latest faulting or fluid flow. Interestingly, the authors identified segments having two similar consecutive ages ("plateaus") in the age-size plot, distinctively different from previously reported examples from other fault gouges. From this relationship, the authors claimed that the "plateaus" near the finest size fractions are generated due to more significant authigenesis, further supporting the conclusion that the finest size batches yield the most reliable K/Ar ages corresponding to authigenesis (presumably during faulting or fluid flow). I believe this manuscript successfully demonstrates how the age-size plot and structural analysis can be used to constrain the timing of authigenesis with higher confidence. However, this manuscript requires further discussion before consideration of publication, as described below.

1. The primary concern of this manuscript is whether or not the K/Ar system of the sample has been completely "closed" since the illite crystallization. There are at least two geological processes that can disturb the original K/Ar system: (1) thermal event(s) that can cause Ar diffusion, and (2) fluid flow that can lead to chemical exchange. The finest-grain batches are used in this study to constrain the most important temporal constraints, but the finest grains are also most prone to Ar diffusion or chemical exchange. Furthermore, the three samples used in this study are collected from a small area - sample locations of the BO-GVI-1 and BO-GVI-2 are only a few tens of cm apart in the same outcrop (Fig. 2b), and sample BO-OFR-1 is also from a nearby outcrop. The authors claimed that the area has experienced multiple geological events of faulting and fluid flow as illustrated in Fig. 5. In such a circumstance, the later geological events may have caused Ar loss of pre-existing illite crystals at the time of each event. It is unclear why the ~200 Ma event caused illite neocrystallization in BO-GVI-1, but did not affect the K/Ar system of the nearby BO-GVI-2. The fine-grain K/Ar ages from the three samples (260 Ma, 200 Ma, 125 Ma) may represent partially reset ages. Combining the biotite Ar diffusion parameters and cylinder geometry used by Grove and Harrison (1996, Am Mineral), variable amounts of Ar loss can occur for different conditions as can be seen in the following examples.

Heating T: Heating duration: Domain size: Ar loss

200C: 1 Myr: 0.1 um: 100%

150C: 1 Myr: 0.1 um: 14%

100C: 1 Myr: 0.1 um: 0%

These Ar loss estimates are likely minimal because (1) the thickness of illite crystals in the finest batch is ~ 0.1 um (Fig. 3a and b), therefore the diffusion domain is probably even smaller than this, and (2) the Ar diffusion in illite is probably faster than in biotite. These Ar loss estimates are not negligible, and the interpretation can be largely dependent on the model.

2. It is claimed that the plateau in the size-age diagram is caused by "significant illite authigenesis" (line 290, lines 374-382). It seems reasonable, but the plateau can be also explained by similar mixing ratios between neocrystalline and protolithic components in different size fractions. Solum et al. (2005, *J Struct Geol*) found that the most intensely deformed fault gouge yielded higher 1Md illite (neocrystalline component)/2M1 illite (protolithic component) value compared to their less intensely deformed samples. Is there any independent evidence suggesting that the Permian event (forming a fine-grain plateau of curve 1, and coarse -grain plateaus of curve 2 and 3) is stronger than the 200 Ma event (not forming a fine-grain plateau of curve 2)?

An alternative interpretation of the plateau can be deduced from two seemingly-opposite processes: (1) neocrystallization of illite and (2) partial resetting of K/Ar system of existing illite. For the neocrystallization through Ostwald ripening, the nucleating particles may have stopped growing (therefore the grain remained small) yielding older K/Ar ages whereas the larger grains that experienced more extended growth would yield younger ages (Clauer, 2013, *Chem Geol*). In this case, a negative age-size relationship is expected. The opposite trend (positive age-size relationship) is commonly observed in many fault gouge samples presumably due to (1) mixing with more protolithic grains as the grain size increases, or (2) partial Ar loss of the existing grains. Therefore, these two competing components will determine the final age-size relationship. Is it possible that the age plateau forms because these two opposite trends cancel each other out for the corresponding size fractions?

Also, it is unclear how the old "plateaus" for the large size fractions can be explained. For the curve 1, why is the protolithic component (350 Ma corresponding to old plateau) is so strong if significant authigenesis occurred during the young plateau age (e.g., 260 Ma)?

3. The described faulting and fluid flow occurred at shallow crustal section during a relatively long period of 260 Ma \sim 125 Ma. How are these events related to the exhumation of the regional area? Are there any low-T thermochronology data? Relating the ambient T conditions to the faulting or fluid flow can provide more detailed depth information of such geological processes.

Kyle Min
University of Florida

Reviewer #2 (Remarks to the Author):

The paper by Viola et al. describes the potential of careful K-Ar dating of authigenic Illite (coupled with detailed field analysis) to unravel the complex temporal history of fault reactivation. They present a clear picture of the internal organization of the Goddo Fault in Western Norway (not previously documented) and solid analytical results documenting three phases of tectonic activity and structural overprinting, including a Cretaceous extensional reactivation not previously recognized (or dated) in the tectonic history of the mid-Norwegian margin.

The Authors build the interpretation of their data on the Age Attractor model presented in Torgersen et al., 2014 (by the same research groups). They expand and further clarify the AA model, clearly

supporting it with new data, which were lacking/unclear in the previous paper.

I think that the new data (supporting both the AA model and a new tectonic information about the study area) and the multidisciplinary approach of the manuscript represent a significant advance in the field of structural geology and geochronology applied to faults. Moreover the paper is well written and the scientific message clear. Therefore I suggest that the paper is accepted for publication in Nature Communications pending (very) minor revisions.

I have only one major observation/request about the science of the paper:

Coarse grain size fraction could be juxtaposed to wallrock ages to better constrain if they represent "true" ages of a tectonic event or are part of the mixing line between the wallrock and the Age Attractor. Especially for BO-GVI-2 the plateau may be apparent and, in addition, there is no clear geologic event that could be related to those ages, with the exception of ref. 47. The same applies to the other two samples, which however both overlap in time (also with the AA of BO-GVI-2) and may be correlated to geologic events that are better constrained. The Authors may show ages of the wallrock they already have (e.g. from Scheiber et al., in prep.) or from the literature.

I have other minor comments, which are not of fundamental importance to deliver the message of the paper, but I hope can help clarifying some points of the manuscript.

MINOR COMMENTS

Line 42: delete "once formed" and I would substitute "are" sensitive with "may be" sensitive. It is a subtle change, but not all faults are reactivated/alterd by far field stresses. Also, a couple of citations backing up the first two sentences are welcome if the manuscript space allows them.

Line 50: if word "saving" is needed, you may delete "fluid infiltration and".

Lines 69-73: I might argue here that recognizing and dating different deformation events is very important but does not "allow" the identification and understanding of deformation processes in fault rocks. Another (very minor) point is the use of the term "brittle". I don't think it is needed as the methodology proposed in this paper may apply to any authigenic/synkinematic Illite, which may occur also in "ductile" faulting processes.

You can simply say (as you do in the following sentence) that you apply the methodology to a brittle reactivation, but you do not need to limit the introductory sentence (l. 69-73) to localized faulting. Just deleting "brittle" in these lines would do the trick in my opinion.

L90-92: K-Ar ages DO decrease with the grain size. I think you could either mention here the presence of "pseudo-plateaus" of the Age Attractor model or delete the sentence, which is not immediately clear to the reader. The sentence is already long and the details of the results are clearly explained in the rest of the manuscript.

L139: subvertical fractures or sub-horizontal striations?

L156: There is (or is recognizable) a slip discontinuity between these two fault rocks?

L171: Mention where the analyses were carried out and which instrument was used iff the manuscript length allows it.

L188 "in which" instead of "where".

L199: from which sample the biotite in 3e comes from? BO-OFR1 ? Write it in the figure caption

L210-213: Again, I do not understand why it is useful discussing here about strain hardening/strain weakening mechanisms. It is not the topic of this manuscript, and, in my opinion, diverts the attention of the reader from the key point of the paper (a methodology that allows to unravel the complex timing of the deformation).

I would just delete this sentence and start the paragraph with the following sentence " The dynamic nature of faults...". I would only keep, if necessary the mention about the short-lived nature of brittle faulting. The very important statement is from L 221 to 228 !

L215: "(e.g. 17)"

L306: could the Authors comment (here in the rebuttal letter) why they refer to the initial faulting phase as "brittle-ductile"?

I understand that the first faulting may occur at higher temperature and with the development of a body of foliated fault rocks, but the faulting looks very localized to me. In addition the Authors mention that the foliation is likely dilatant, as demonstrated by the occurrence of foliation-parallel veins. Localization and dilatant (i.e. pressure-sensitive) deformation are typical of brittle faulting. Do you have evidence of dynamic recrystallization within the phyllonite and/or the hangingwall to support a ductile phase? The phyllonite could be the result of chemical alteration and/or pressure solution in the "brittle" regime, as in the case of other documented examples like the faults in Torgersen et al., 2014, the Zuccale fault, the San Andreas and many others... In any case, this point does not influence the present manuscript in a significant way.

L320-323: I am a bit confused here. It is indeed possible that a fault mirror is created by fast fault slip (although some may argue that they also may form under subseismic slip velocities, e.g. Verbeke et al., 2013, *Geology*), but I would cite Kirkpatrick et al., 2013, *Geology* since they describe quartz-based fault mirrors, instead of Dolostone mirrors as in Fondriest et al., 2013 (or cite both). What puzzles me is the mention to an additional brittle faulting phase and about contraction. Brittle faulting is typically characterized by localization and the formation of the PSS, synthetic Riedels and the cataclasites may well be contemporaneous. Secondly, why shortening? Isn't it the normal faulting cross-cutting older transpressive structures (lines 134-136) ?

L391-393: same comment as above. In my opinion the simple presence of a phyllonite does not mean a ductile shear zone, not in the same sense of a mylonitic shear zone, anyway.

L615: I guess it is (f) instead of (d).

Telemaco Tesei
INGV Rome
telemaco.tesei@ingv.it

Reviewer #3 (Remarks to the Author):

Review: "Deconvoluting complex structural histories archived in brittle fault zones" by Viola et al.

The authors apply K-Ar dating to illite-rich fault gouge samples to unravel the complex history of the Goddo Fault in SW Norway. Dating of multiple grain size fractions from several samples from the same fault, coupled with detailed structural observations allow the authors to constrain a sequence of geologic events spanning more than 200 myr and including three periods of fault activity or reactivation in Permian, late Triassic-early Jurassic and mid Cretaceous times respectively.

This is an excellent regional study and as such of great interest to the geological community working in the Norway/North Sea/North Atlantic region. The dataset and interpretations are sound, original and well-presented and very relevant to the tectonic history of the Norwegian margin.

From a methodological point of view, this study is not entirely a novel approach. The method has been around since the 1970s (Lyons and Snellenburg, 1971) and dozens of papers have been published in the last two decades (see references in manuscript and many others). Other authors have to some degree integrated fault geochronology with structural studies (e.g. Vrolijk and van der Pluijm, 1999; Solum et al., 2005; Haines and van der Pluijm, 2010; Davids et al., 2013 and others), including publications that were (co-)authored by Viola (Viola et al., 2013; Torgersen et al., 2014; 2015). However, the thoroughness and detail in linking structural observations with detailed geochronology to really understand in depth the history of one multiply reactivated fault are commendable and should be the standard for future publications in the field. This is definitely a step toward a better understanding of fault behaviour and timing. In this regard, the manuscript could become a landmark paper on how to present and interpret K-Ar fault rock geochronology and unravel complex fault systems. This is the main strength and should be the focus of the manuscript. However, at the moment, it reads a little as if the authors are desperately trying to sell their approach as innovative and novel, when it is really more of an improvement and development of pre-existing methods.

Overall this is an excellent contribution that could be published with very little additional work.

Minor comments:

Introduction:

According to Nature Communication's 'Guide to authors', the introduction should be a "referenced text that expands on the background of the work". At the moment, the introduction reads like an extended abstract, summarising also the methods, results and conclusions (page 4-5, lines 68-83 and 88-102). These parts could be shortened or moved to the respective later sections altogether, to make room for the geological setting that is now part of the results chapter.

Page 3, lines 56-60: If this is indeed "generally acknowledged", please provide some additional references that are not (co-)authored by Viola.

Page 4, line 80: 'the fault's' (not 'the fault')

Results:

Pages 5-6, lines 106-123: Geological setting should not be part of the results chapter - move to the introduction.

Page 5, line 111: Delete 'mountain belt'

Page 8, line 176: Why use an out-dated timescale and not the more recent Gradstein et al. (2012) or even the stratigraphic chart from 2016 (<http://www.stratigraphy.org/index.php/ics-chart-timescale>)?

Page 8, line 189: Feldspar wherein? The last sentence states that there is no feldspar in the finest fraction, so is this in the coarser fractions of sample BO-GVI-1?

Discussion:

At the moment, the mineralogy and polytypes are not discussed, even though they have been determined and including them could improve the interpretation and/or add to the understanding of illite/mica growth in faults. The coarse fractions of sample BO-GVI-2 for example contain both 2M1 illite/mica (32-33 %), interstratified illite/smectite (31-35 %) and dioctahedral smectite (16-17 %) which grow normally at different temperatures. In the Devonian-early Carboniferous, temperatures in the basement rocks should have been much too high for smectite to be stable. Do you think the smectite and 2M1 illite/mica grew contemporaneously and if so, why? Or could the smectite indicate later re-activation? In the Triassic-Jurassic it seems to be predominantly interstratified illite/smectite that is growing and in the mid-Cretaceous just smectite (although the Permian coarse fractions from sample BO-OFR-1 also contain mainly smectite). Generally, this seems to record cooling of the basement host rocks. Surely, the mineralogy and polytypes deserve some attention?

Page 10, line 233: 'detrital' normally refers to material derived by weathering and erosion, so should not be used in this context, i.e. when referring to illite/mica from crystalline host rocks or from earlier fault activity. 'Inherited' might be a better term.

Page 13, line 304: 'the data suggest' (not 'suggests')

Conclusions:

Conclusions could be more succinct and to the point.

Page 16, lines 368-369: This should be mentioned before coming to the conclusions.

Page 16, line 375: replace 'by' with 'with'; comma after 'unusual'

Page 16, lines 383-387: This is not the first study using K-Ar dating of illite (and/or K-feldspar) to date fault-related fluid flow and alteration (e.g. Siebel et al., 2010; Brockamp and Clauer, 2013).

Figures:

Generally, very nice figures.

Fig. 1b: mark location of sampled outcrop.

Page 23, line 621: Not necessarily the last faulting episode, just the last episode recorded in the illite data. Several studies indicate that faults can move without crystallising or resetting illite (e.g. Parry et al., 2001; Duvall et al., 2011; Sasseville et al., 2012).

References:

Generally, the choice of references seems to strongly favour publications coming from European and Australian laboratories. The dozens of papers from North American authors on K-Ar and Ar-Ar fault dating are somewhat underrepresented.

Writing style:

While the manuscript is generally very well written, the style is somewhat cumbersome to read, with long and complex sentences. The authors could get their points across much better and make reading more enjoyable (especially for non-native speakers) with shorter sentences.

Reviewer: A. Ksienzyk

Reviewer #1 (Remarks to the Author):

Review of the manuscript "Deconvoluting complex structural histories archived in brittle fault zones", by Viola et al.

The pioneering work of van der Pluijm et al. (2001, Nature), who found a general trend in Ar/Ar age - 2M_d/1M (detrital illite %) plot, provided a fundamental basis for constraining the timing of latest illite authigenesis. Their primary argument is that each fault gouge sample is composed of two endmembers: (1) a protolithic component whose K/Ar system is not affected by authigenesis, and (2) an authigenic component formed during late stage fault activity. Based on the positive, linear relationship in age - detrital illite %, they concluded that the y-intercept represents the timing of illite authigenesis, presumably during the latest fault activity. In later works, the Michigan group (Solum et al., 2005, J Struc Geol) developed more convincing evidence showing two linear trends with differential slopes, but identical y-intercept in the age - detrital illite % plot, reinforcing the previous conclusion that the finest size fraction yields the timing of latest authigenesis. Similar or slightly modified approaches have been applied for fault gouge samples.

This manuscript provides detailed structural data and K/Ar ages to constrain timing of faulting and fluid flow that occurred at a relatively shallow crustal section. Based on the general increase of K/Ar ages with size fractions, it was concluded that the finest size fractions yield the most reliable K/Ar ages corresponding to the timing of the latest faulting or fluid flow. Interestingly, the authors identified segments having two similar consecutive ages ("plateaus") in the age-size plot, distinctively different from previously reported examples from other fault gouges. From this relationship, the authors claimed that the "plateaus" near the finest size fractions are generated due to more significant authigenesis, further supporting the conclusion that the finest size batches yield the most reliable K/Ar ages corresponding to authigenesis (presumably during faulting or fluid flow). I believe this manuscript successfully demonstrates how the age-size plot and structural analysis can be used to constrain the timing of authigenesis with higher confidence. However, this manuscript requires further discussion before consideration of publication, as described below.

1. The primary concern of this manuscript is whether or not the K/Ar system of the sample has been completely "closed" since the illite crystallization. There are at least two geological processes that can disturb the original K/Ar system: (1) thermal event(s) that can cause Ar diffusion, and (2) fluid flow that can lead to chemical exchange. The finest-grain batches are used in this study to constrain the most important temporal constraints, but the finest grains are also most prone to Ar diffusion or chemical exchange. Furthermore, the three samples used in this study are collected from a small area - sample locations of the BO-GVI-1 and BO-GVI-2 are only a few tens of cm apart in the same outcrop (Fig. 2b), and sample BO-OFR-1 is also from a nearby outcrop. The authors claimed that the area has experienced multiple geological events of faulting and fluid flow as illustrated in Fig. 5. In such a circumstance, the later geological events may have caused Ar loss of pre-existing illite crystals at the time of each event. It is unclear why the ~200 Ma event caused illite neocrystallization in BO-GVI-1, but did not affect the K/Ar system of the nearby BO-GVI-2. The fine-grain K/Ar ages from the three samples (260 Ma, 200 Ma, 125 Ma) may represent partially reset ages. Combining the biotite Ar diffusion parameters and cylinder geometry used by Grove and Harrison (1996, Am Mineral), variable amounts of Ar loss can occur for different conditions as can be seen in the following examples.

Heating T: Heating duration: Domain size: Ar loss

200C: 1 Myr: 0.1 um: 100%

150C: 1 Myr: 0.1 μm : 14%

100C: 1 Myr: 0.1 μm : 0%

These Ar loss estimates are likely minimal because (1) the thickness of illite crystals in the finest batch is $\sim 0.1 \mu\text{m}$ (Fig. 3a and b), therefore the diffusion domain is probably even smaller than this, and (2) the Ar diffusion in illite is probably faster than in biotite. These Ar loss estimates are not negligible, and the interpretation can be largely dependent on the model.

The reviewer raises an important issue that is often debated in the Ar/Ar community. It has already been in part dealt with by some of us in the past and our take is available in published peer-reviewed papers (e.g. Zwingmann et al., 2010 and Torgersen et al., 2014). In summary, as discussed in more detail here below, we do not think that radiogenic ^{40}Ar volume diffusion can affect significantly the age-grain size spectra.

*Dating assumes that no isotopic re-equilibration has occurred since the dated minerals formed. However, as pointed out by the reviewer, exposure to temperatures at or above the formation temperature for **considerable time intervals** may cause volume diffusion of radiogenic ^{40}Ar and thus a partial or full reset of the system, leading to mixed ages.*

A first comment deals with the peculiar nature of fault zones, whose transient behaviour (both mechanical and thermal) represents a significant difference to the “static” environments that are generally used when conceptualizing Ar diffusion in coarse mica grains.

*The effects of thermally-induced volume diffusion can be best explored by a numerical modelling approach. Unfortunately, however, Ar diffusion modelling in clays is not common, due to conceptual difficulties arising from their fine grain size and poorly constrained diffusion parameters. Thus, a simple transfer of modeling results obtained for relatively coarse mineral grains (such as biotite or muscovite $> 100 \mu\text{m}$) during long-lasting and gradual cooling following regional metamorphism, as proposed by the reviewer, is certainly not ideal to clays formed **within short-lived faults with only transient high temperatures and important fluid-rock interaction**.*

Potential partial Argon resetting within clay-size crystallites has been previously done with an Argon diffusion code by, for example, Zwingmann et al. (2010), based on the diffusion model by Huon et al. (1993), and Torgersen et al. (2014), based instead on the script DIFFARG by Wheeler (1996) implemented with the most recent diffusion parameters by Harrison et al. (2009).

A potential thermal overprint on the ages of the illite fractions of our study was evaluated by the Ar diffusion calculations illustrated in Fig. 1, based on the parameters by Huon et al. (1993).

Figure 1: Radiogenic ⁴⁰Ar loss for cylindrical (CYL) or platy (PLA) geometries of a 0.1 μm grain size clay crystallite as a function of exposure times of up to 1Myr to temperatures of 100, 150 and 200° C.

Given the conceptual importance of the finest grain size in our study (which, we conclude, is the fraction whose age comes the closest to the age of the last recorded faulting episode), a maximum grain sizes of 0.1 μm was used within a temperature range from 100 to 200 °C and a timeframe of 1 Ma. The chosen temperature interval is believed to be well representative based on the total lack of ductile deformation in the region, existing thermochronology data (which are now explicitly referred to and discussed in the paper) and on our own, still unpublished, clumped isotope results produced from synkinematic carbonates within many faults of the area. The short time frame is probably not meaningful in the case of regional thermal events, but is appropriate for representing the life span of single faulting episodes and fault ruptures with associated fluid ingress. As the geometry of fine grained clay minerals (fibres, plates) has a strong influence on Ar diffusion, both cylindrical and plane geometric shapes were considered. The original parameters listed in Huon et al. (1993) were used in this test with D_0 and E_a values $6.03 \times 10^{-7} \text{ cm}^2/\text{s}$ and $40 \times 10^3 \text{ cal/mol}$ respectively (Wijbrans and McDougall, 1986). We prefer to use these values and model for a diffusion calculation compared to the suggested biotite model by the reviewer.

The Ar diffusion modelling (Fig. 1) reveals that temperatures of up to 150 °C are negligible for diffusion within a cylinder and plate shape particle with 0.1 μm grain size. Instead, up to 42 % of radiogenic Ar could be lost by diffusion if the overprinting temperature reached 200° C for a 0.1 μm cylindrical clay particle and up to 22 % for a plate shape particle with the same grain size within a 1 Ma overprint time (Fig. 1). As no systematic shift or disturbance was observed in the presented age data, we consider Ar loss by diffusion an unlikely process in the case of the Goddo fault.

Even more convincing are the results that we can extrapolate from the study of Torgersen et al. (2014; Fig.2; I am author nr. 2 in that study). Their calculations assumed a cylindrical grain geometry and modeling was repeated for a range of grain-sizes (10, 2 and 0.1 μm), peak temperatures (190-

370°C) and duration of thermal episodes (5 and 10 Ma). As shown in Fig. 2, they concluded that during heating-cooling pulses of 5 and 10 Ma (so, longer than the 1 Ma case used in the modeling of fig. 1) to temperatures of 230-240°C, even very fine-grained illites (< 0.1 µm) do not experience more than a 10% resetting of their initial K/Ar age. Only at 300-310°C, the Ar isotopic system of the < 0.1 and 2 µm grains is completely reset.

Fig. 4. Results of Ar diffusion modeling, with modeled age and percentage of radiogenic Ar loss plotted against peak temperature. The diagram shows the impact on 800 Ma 1/Ms (grain sizes: 0.1, 2 and 10 µm) of transient elevated temperatures peaking at 418 Ma. Color-coding reflects the different grain sizes, whereas solid and striped lines mark 5 and 10 Ma long thermal episodes, respectively. The red inset diagram shows the modeled temperature curves, in which solid and striped lines correspond to those in the main diagram. (For interpretation of the references to color in this figure legend, the reader is referred to the web version of this article.)

Figure 2: From Torgersen et al. (2014)

In summary, although Ar diffusion cannot and should not be completely ruled out, we feel confident that our internally consistent data do not suggest a significant influence of radiogenic ⁴⁰Ar diffusion and this is well supported by diffusion modelling done with the most recent and relevant diffusion parameters for clays.

The implications of this conclusion are indeed at the core of our study, which confirms the dramatic compartmentalization of the studied fault not only from a geometric/kinematic and mechanic perspective, but also from a thermal and hydrological point of view.

We have expanded the text in the paper to clarify this point.

2. It is claimed that the plateau in the size-age diagram is caused by "significant illite authigenesis" (line 290, lines 374-382). It seems reasonable, but the plateau can be also explained by similar mixing ratios between neocrystalline and protolithic components in different size fractions. Solum et al. (2005, J Struc Geol) found that the most intensely deformed fault gouge yielded higher 1Md illite (neocrystalline component)/2M1 illite (protolithic component) value compared to their less intensely deformed samples. Is there any independent evidence suggesting that the Permian event (forming a fine-grain plateau of curve 1, and coarse -grain plateaus of curve 2 and 3) is stronger than the 200 Ma event (not forming a fine-grain plateau of curve 2)?

The Permian is known as "THE" thermal event in southern Norway, causing the major Oslo Rift and much of the early tectonothermal activity in the Viking Graben. Although just at a speculative level, it seems reasonable to expect a more significant thermal overprint of Permian age. And our data confirm this, with authigenesis of clays with Permian ages all the way to the 10 µm grain size.

As for the suggested possibility of similar mixing ratios: it is indeed a possibility. However, it seems highly unlikely that this would occur only for the three samples of this study and for their different grain size fractions, but has never been reported elsewhere in other studies. What would be the reason as to why there are 4 “plateaus” in our study if this only reflected mixing? It strikes as an incredible coincidence if it were true.

An alternative interpretation of the plateau can be deduced from two seemingly-opposite processes: (1) neocrystallization of illite and (2) partial resetting of K/Ar system of existing illite. For the neocrystallization through Ostwald ripening, the nucleating particles may have stopped growing (therefore the grain remained small) yielding older K/Ar ages whereas the larger grains that experienced more extended growth would yield younger ages (Clauer, 2013, Chem Geol). In this case, a negative age-size relationship is expected. The opposite trend (positive age-size relationship) is commonly observed in many fault gouge samples presumably due to (1) mixing with more protolithic grains as the grain size increases, or (2) partial Ar loss of the existing grains. Therefore, these two competing components will determine the final age-size relationship. Is it possible that the age plateau forms because these two opposite trends cancel each other out for the corresponding size fractions?

Our last comment above applies to this possibility as well. Moreover, a negative age-size relationship, although theoretically indeed possible, is extremely rare and we have personally documented it in no more than a couple of cases in our career (out of the hundreds of individual fault rock K-Ar spectra produced). There is no sign of inversion in our dataset and we therefore prefer to believe that the interpretation given in the text is the most likely and is certainly scientifically sound and plausible.

Also, it is unclear how the old “plateaus” for the large size fractions can be explained. For the curve 1, why is the protolithic component (350 Ma corresponding to old plateau) is so strong if significant authigenesis occurred during the young plateau age (e.g., 260 Ma)?

Had authigenesis been complete, then the plateaus for the large size fractions would not exist, and a flat age vs grain size diagram would be expected aligned with the age of the finest. Although rare, this situation has indeed been documented already by, for example, Torgersen et al. (2014), whose sample ETO_040 (their Fig. 6) has four fractions all with the same age.

There exists, therefore, a very delicate balance between authigenesis (starting from the finest fraction sizes and propagating progressively toward the coarser fractions as more and more newly crystallized illites form and grow in size) and the inherited protolithic component of the coarse fractions. The documented patterns of our study are new in the literature and help indeed bridge a conceptual gap between inclined spectra (wherein each fraction has a different age) and the ideal situation of sample ETO_040 of Torgersen et al. (2014).

3. The described faulting and fluid flow occurred at shallow crustal section during a relatively long period of 260 Ma ~ 125 Ma. How are these events related to the exhumation of the regional area? Are there any low-T thermochronology data? Relating the ambient T conditions to the faulting or fluid flow can provide more detailed depth information of such geological processes.

This is an interesting point that is now better addressed in the revised version. We provide a reference to existing geochronological data and a recently published paper by our group (Scheiber et al., in press, Terra Nova) that offers also a regional perspective on existing cooling ages and direct dating of deformation for the late Caledonian time period.

Kyle Min
University of Florida

Reviewer #2 (Remarks to the Author):

The paper by Viola et al. describes the potential of careful K-Ar dating of authigenic Illite (coupled with detailed field analysis) to unravel the complex temporal history of fault reactivation. They present a clear picture of the internal organization of the Goddo Fault in Western Norway (not previously documented) and solid analytical results documenting three phases of tectonic activity and structural overprinting, including a Cretaceous extensional reactivation not previously recognized (or dated) in the tectonic history of the mid-Norwegian margin.

The Authors build the interpretation of their data on the Age Attractor model presented in Torgersen et al., 2014 (by the same research groups). They expand and further clarify the AA model, clearly supporting it with new data, which were lacking/unclear in the previous paper.

I think that the new data (supporting both the AA model and a new tectonic information about the study area) and the multidisciplinary approach of the manuscript represent a significant advance in the field of structural geology and geochronology applied to faults. Moreover the paper is well written and the scientific message clear. Therefore I suggest that the paper is accepted for publication in Nature Communications pending (very) minor revisions.

I have only one major observation/request about the science of the paper:

Coarse grain size fraction could be juxtaposed to wallrock ages to better constrain if they represent "true" ages of a tectonic event or are part of the mixing line between the wallrock and the Age Attractor. Especially for BO-GVI-2 the plateau may be apparent and, in addition, there is no clear geologic event that could be related to those ages, with the exception of ref. 47. The same applies to the other two samples, which however both overlap in time (also with the AA of BO-GVI-2) and may be correlated to geologic events that are better constrained. The Authors may show ages of the wallrock they already have (e.g. from Scheiber et al., in prep.) or from the literature.

Our paper can now draw from the contribution by Scheiber et al., which is presently officially in press. This permits to better elaborate on the significance of the early cooling events of the host granodiorite and on the exact timing of the brittle deformation events that affected the region in the Late Ordovician and Silurian. The thermal imprint of those episodes is obviously the isotopic starting point for the later evolution. This is now better described in the amended version of the paper.

In addition, we have added new text and specific references referring the reader to the available lower-T thermochronological studies (apatite fission track and (U-Th)/He) for the study area and surrounding regions. An important fact that now emerges clearly in the amended text is that the regional temperatures did not exceed 60° C from at least the early Jurassic. This lends further credibility to the Age Attractor approach that we refine in the study and allows us to exclude significant Ar volume diffusion as a plausible cause for the age vs. grain size spectra of our samples.

I have other minor comments, which are not of fundamental importance to deliver the message of

the paper, but I hope can help clarifying some points of the manuscript.

MINOR COMMENTS

Line 42: delete "once formed" and I would substitute "are" sensitive with "may be" sensitive. It is a subtle change, but not all faults are reactivated/altered by far field stresses. Also, a couple of citations backing up the first two sentences are welcome if the manuscript space allows them.

We prefer to not delete "once formed" as this helps to stress the time that is involved between the initial fault nucleation and the later reactivation. "Are" has been changed" to "may be", as requested.

Line 50: if word "saving" is needed, you may delete "fluid infiltration and".

"fluid infiltration and hence" are now removed.

Lines 69-73: I might argue here that recognizing and dating different deformation events is very important but does not "allow" the identification and understanding of deformation processes in fault rocks.

We prefer to stick to the initial version. The observation of the reviewer neglects the first part of our sentence, stating that our approach first of all allows the construction of "time-constrained" brittle evolutionary models. Only at a second stage, and in turn, this leads to the investigation and understanding of the rest. This is exactly our point: without good control on the time-constrained evolution of a fault, inferences on deformation mechanisms may simply be wrong! A minor change to the text was nonetheless implemented.

Another (very minor) point is the use of the term "brittle". I don't think it is needed as the methodology proposed in this paper may apply to any authigenic/synkinematic Illite, which may occur also in "ductile" faulting processes.

No, this is not true. K-Ar dating of authigenic illite can only be applied to K-bearing clay minerals (illite), which are stable exclusively at T conditions leading to brittle deformation. Illite will not remain stable under thermal conditions leading to viscous deformation. Other geochronological methodologies will have to be used to date mylonitization processes.

You can simply say (as you do in the following sentence) that you apply the methodology to a brittle reactivation, but you do not need to limit the introductory sentence (l. 69-73) to localized faulting. Just deleting "brittle" in these lines would do the trick in my opinion.

See last comment.

L90-92: K-Ar ages DO decrease with the grain size. I think you could either mention here the presence of "pseudo-plateaus" of the Age Attractor model or delete the sentence, which is not immediately clear to the reader. The sentence is already long and the details of the results are clearly explained in the rest of the manuscript.

Point taken. The sentence has been deleted.

L139: subvertical fractures or sub-horizontal striations?

True. The text was not correct and also slightly confusing. It has now been improved.

L171: Mention where the analyses were carried out and which instrument was used iff the manuscript length allows it.

Done

L188 "in which" instead of "where".

Done.

L199: from which sample the biotite in 3e comes from? BO-OFR1 ? Write it in the figure caption

Done.

L210-213: Again, I do not understand why it is useful discussing here about strain hardening/strain weakening mechanisms. It is not the topic of this manuscript, and, in my opinion, diverts the attention of the reader from the key point of the paper (a methodology that allows to unravel the complex timing of the deformation).

I would just delete this sentence and start the paragraph with the following sentence " The dynamic nature of faults...". I would only keep, if necessary the mention about the short-lived nature of brittle faulting. The very important statement is from L 221 to 228 !

We followed the suggestion and restructured the sentence such that it now starts with "The dynamic nature...". We also decided to drop the reference to strain hardening and weakening.

L215: "(e.g. 17)"

Done.

L306: could the Authors comment (here in the rebuttal letter) why they refer to the initial faulting phase as "brittle-ductile"?

I understand that the first faulting may occur at higher temperature and with the development of a body of foliated fault rocks, but the faulting looks very localized to me. In addition the Authors mention that the foliation is likely dilatant, as demonstrated by the occurrence of foliation-parallel veins. Localization and dilatant (i.e. pressure-sensitive) deformation are typical of brittle faulting. Do you have evidence of dynamic recrystallization within the phyllonite and/or the hangingwall to support a ductile phase? The phyllonite could be the result of chemical alteration and/or pressure solution in the "brittle" regime, as in the case of other documented examples like the faults in Torgersen et al., 2014, the Zuccale fault, the San Andreas and many others... In any case, this point does not influence the present manuscript in a significant way.

Localised vs. not localised is a scale-dependent concept. In this study we do not provide too much geological background to the study and it would become therefore too difficult to argue in detail as to why there might be a phase of initial brittle-ductile localization. A convincing piece of evidence is, however, the increased amount of phyllosilicates in the foliated core of the fault, possibly the result of strain accommodation under T conditions that are well in line with those required in a granodiorite to form mica.

To argue against the view of the reviewer, moreover, one could suggest that the dilatant phase of deformation, responsible for the foliation-parallel veins, is the result of a later, pure brittle and dilatant reactivation of the previously formed foliation.

This is in itself a clear example of the main message of our paper that the reviewer perhaps does not see right away. Structural features that are currently coexisting and are spatially associated, are not necessarily the result of the same deformation episode. They may simply reflect the finite summation of deformation episodes (of different age) that, through a process of structural reactivation, affect the very same rock volume.

However, in order to not distract the reader from the main thrust of the paper (and its main message), we decided to simplify it throughout by removing (as suggested) any reference to a possible transition from a brittle-ductile stage to a fully brittle one.

L320-323: I am a bit confused here. It is indeed possible that a fault mirror is created by fast fault slip (although some may argue that they also may form under subseismic slip velocities, e.g. Verbeke et al., 2013, Geology), but I would cite Kirkpatrick et al., 2013, Geology since they describe quartz-based fault mirrors, instead of Dolostone mirrors as in Fondriest et al., 2013 (or cite both).

Done, both references are now cited.

What puzzles me is the mention to an additional brittle faulting phase and about contraction. Brittle faulting is typically characterized by localization and the formation of the PSS, synthetic Riedels and the cataclasites may well be contemporaneous. Secondly, why shortening? Isn't it the normal faulting cross-cutting older transpressive structures (lines 134-136)?

Our mistake. It is not shortening, it is extensional deformation. Text is now corrected.

The “additional” phase of deformation does not need to be that distinct in time (again, the time dimension). What does “contemporaneous” mean to the reviewer? All we can see is a simple geometric cross-cutting relationship, which is a clear indication of the “younger” age of the feature that cross-cuts. So, in summary, the meaning of the term “renewed” (which is now used in the revised version) is that of expressing a younger age of the PSS in comparison to the gouge, without wanting to imply, however, a distinct tectonic event!

L391-393: same comment as above. In my opinion the simple presence of a phyllonite does not mean a ductile shear zone, not in the same sense of a mylonitic shear zone, anyway.

See the comment above for line 306. Again, we have now removed the mention to a progressive transition from brittle-ductile to brittle.

L615: I guess it is (f) instead of (d).

Yes, text now corrected to (f).

Telemaco Tesei
INGV Rome
telemaco.tesei@ingv.it

Reviewer #3 (Remarks to the Author):

Review: "Deconvoluting complex structural histories archived in brittle fault zones" by Viola et al.

The authors apply K-Ar dating to illite-rich fault gouge samples to unravel the complex history of the Goddo Fault in SW Norway. Dating of multiple grain size fractions from several samples from the same fault, coupled with detailed structural observations allow the authors to constrain a sequence of geologic events spanning more than 200 myr and including three periods of fault activity or reactivation in Permian, late Triassic-early Jurassic and mid Cretaceous times respectively.

This is an excellent regional study and as such of great interest to the geological community working in the Norway/North Sea/North Atlantic region. The dataset and interpretations are sound, original and well-presented and very relevant to the tectonic history of the Norwegian margin.

From a methodological point of view, this study is not entirely a novel approach. The method has been around since the 1970s (Lyons and Snellenburg, 1971) and dozens of papers have been published in the last two decades (see references in manuscript and many others). Other authors have to some degree integrated fault geochronology with structural studies (e.g. Vrolijk and van der Pluijm, 1999; Solum et al., 2005; Haines and van der Pluijm, 2010; Davids et al., 2013 and others), including publications that were (co-)authored by Viola (Viola et al., 2013; Torgersen et al., 2014; 2015). However, the thoroughness and detail in linking structural observations with detailed geochronology to really understand in depth the history of one multiply reactivated fault are commendable and should be the standard for future publications in the field. This is definitely a step toward a better understanding of fault behaviour and timing. In this regard, the manuscript could become a landmark paper on how to present and interpret K-Ar fault rock geochronology and unravel complex fault systems. This is the main strength and should be the focus of the manuscript.

We are pleased by and appreciate this comment by the reviewer. We wish to point out that what we believe is entirely novel is not the geochronological component (although we are pushing the K-Ar dating of illite from fault rocks to a new, unprecedented level), nor the structural analysis of brittle faults. The absolute novelty rests in the combined approach, where ONLY an extremely careful structural analysis of complex faults can provide the right background for sampling and dating fault rocks with the goal to unravel complex structural histories.

However, at the moment, it reads a little as if the authors are desperately trying to sell their approach as innovative and novel, when it is really more of an improvement and development of pre-existing methods.

We have now taken onboard this criticism and have toned down several text lines where we were trying to stress (perhaps too vehemently) the novelty of our approach. This notwithstanding, we still believe that, as indicated in the comment immediately above, we offer a fresh and innovative look at this scientific issue by proposing a new approach.

Overall this is an excellent contribution that could be published with very little additional work.

Minor comments:

Introduction:

According to Nature Communication's 'Guide to authors', the introduction should be a "referenced text that expands on the background of the work". At the moment, the introduction reads like an extended abstract, summarising also the methods, results and conclusions (page 4-5, lines 68-83 and 88-102). These parts could be shortened or moved to the respective later sections altogether, to make room for the geological setting that is now part of the results chapter.

We have followed the instructions to authors when structuring the text. Please see also comment below.

Page 3, lines 56-60: If this is indeed "generally acknowledged", please provide some additional references that are not (co-)authored by Viola.

The reviewer is entirely right and the selection of references has now been enlarged.

Page 4, line 80: 'the fault's' (not 'the fault')

Done

Results:

Pages 5-6, lines 106-123: Geological setting should not be part of the results chapter - move to the introduction.

We agree entirely on this point, but it is actually explicitly written in the Instructions to authors that the Geological setting shall be in the results.....

Page 5, line 111: Delete 'mountain belt'

Done

Page 8, line 176: Why use an out-dated timescale and not the more recent Gradstein et al. (2012) or even the stratigraphic chart from 2016 (<http://www.stratigraphy.org/index.php/ics-chart-timescale>)?

Done. Gradstein et al. (2012) is now cited and all the geochronological terminology has now been amended so as to have it fully consistent with the Epochs names reported in it.

Page 8, line 189: Feldspar wherein? The last sentence states that there is no feldspar in the finest fraction, so is this in the coarser fractions of sample BO-GVI-1?

Discussion:

At the moment, the mineralogy and polytypes are not discussed, even though they have been determined and including them could improve the interpretation and/or add to the understanding of illite/mica growth in faults. The coarse fractions of sample BO-GVI-2 for example contain both 2M1 illite/mica (32-33 %), interstratified illite/smectite (31-35 %) and dioctahedral smectite (16-17 %) which grow normally at different temperatures. In the Devonian-early Carboniferous, temperatures in the basement rocks should have been much too high for smectite to be stable. Do you think the smectite and 2M1 illite/mica grew contemporaneously and if so, why? Or could the smectite indicate later re-activation? In the Triassic-Jurassic it seems to be predominantly

interstratified illite/smectite that is growing and in the mid-Cretaceous just smectite (although the Permian coarse fractions from sample BO-OFR-1 also contain mainly smectite). Generally, this seems to record cooling of the basement host rocks. Surely, the mineralogy and polytypes deserve some attention?

A short section has been added with a cautious interpretation of the available XRD data. We agree with the reviewer in the conclusion that the detected mineralogy in the dated samples contains clays that are very likely the result of multiple reactivation of the fault under progressively decreasing temperatures.

Page 10, line 233: 'detrital' normally refers to material derived by weathering and erosion, so should not be used in this context, i.e. when referring to illite/mica from crystalline host rocks or from earlier fault activity. 'Inherited' might be a better term.

Done. We now use inherited.

Page 13, line 304: 'the data suggest' (not 'suggests')

Done.

Conclusions:

Conclusions could be more succinct and to the point.

Page 16, lines 368-369: This should be mentioned before coming to the conclusions.

Page 16, line 375: replace 'by' with 'with'; comma after 'unusual'

We have changed the text but chosen a different solution:are unusual because of the presence.....

Page 16, lines 383-387: This is not the first study using K-Ar dating of illite (and/or K-feldspar) to date fault-related fluid flow and alteration (e.g. Siebel et al., 2010; Brockamp and Clauer, 2013).

The reviewer is correct. New references have been added.

Figures:

Generally, very nice figures.

Fig. 1b: mark location of sampled outcrop.

Done.

Page 23, line 621: Not necessarily the last faulting episode, just the last episode recorded in the illite data. Several studies indicate that faults can move without crystallising or resetting illite (e.g. Parry et al., 2001; Duvall et al., 2011; Sasseville et al., 2012).

OK, we have changed the text to clearly indicated that the finest represents the age of the last faulting episode "recorded by the illite data". This now allows for minor movements and slips that do not lead to any authigenesis in the fault rocks.

References:

Generally, the choice of references seems to strongly favour publications coming from European and Australian laboratories. The dozens of papers from North American authors on K-Ar and Ar-Ar fault dating are somewhat underrepresented.

We agree and we apologize for this. We have introduced several new references and believe that by doing so we offer a more balanced overview of the available literature on the topic.

Writing style:

While the manuscript is generally very well written, the style is somewhat cumbersome to read, with long and complex sentences. The authors could get their points across much better and make reading more enjoyable (especially for non-native speakers) with shorter sentences.

This is a difficult comment to handle, that might also reflect the personal writing style of the reviewer. Given that the other two reviewers found the paper well written and did not complain about it being in place cumbersome, we prefer to refrain from major modifications. Still, we have tried to simplify some of the longest sentences and we hope that this helps.

Reviewer: A. Ksienzyk

REVIEWERS' COMMENTS:

Reviewer #1 (Remarks to the Author):

The revised version of the manuscript properly addressed my original comments, and I believe this manuscript is ready for publication. I agree that plane or cylindrical diffusion geometry is more relevant for clay minerals, but it looks like the new modeling results also suggest that the diffusive Ar loss can be significant (up to 40% loss for isothermal heating at 200C for 1 Myr). Torgersen et al's (2014) model using another type of time-Temperature path (linear T increase followed by linear T decrease) yielded less Ar loss. Although it is an open question whether or not the system was completely closed since last fault activity or fluid flow, I believe it is reasonable to present the modeling results as far as the assumptions are properly laid out. In this regard, the statements in lines 327-333 can be improved either by citing Torgersen et al. (2014) or adding a brief description on the shape of the time-Temperature path used here.

Line 327-333: "Moreover, although radiogenic ^{40}Ar volume diffusion does indeed occur, it is estimated that it does not cause more than a 10% resetting of an K-Ar age during heating pulses up to 230-240°C and lasting up to 10 Ma, even for very fine-grained illites ($< 0.1 \mu\text{m}$). As mentioned earlier, the lack of ductile deformation in the study area and the available thermochronological data (44-46) suggest that the Goddo Fault was never exposed to a temperature higher than c. 250°C."

Other than this, I don't see any other issues. It is a nice piece of work!

Kyle Min
University of Florida

Reviewer #2 (Remarks to the Author):

I am satisfied with the responses the Authors made to my previous revision. They also thoroughly addressed the problem of Ar diffusion raised by another reviewer and therefore I think the paper can be accepted for publication.

Reviewer #3 (Remarks to the Author):

Only two very minor comments:

Page 3, line 67: These seem to be the same three references still (2-4) - see comment and reply on earlier version.

Page 12, line 282: Larsen et al. (2003, NJG 83, 149-165) dated hydrothermally altered feldspar close to faults at 360-370 Ma, suggesting that faults were still active. Slightly older and also further north (Bergen area), but maybe this is useful information.

Looking forward to seeing this published!

In detail, I have done the following changes:

- Reviewer 1:

As suggested by the reviewer, I have improved the statement in lines 327-333 by specifically citing Torgersen et al. (2014). By doing so, the reader is made fully aware of the assumptions and of the boundary conditions of the modelling work available in the literature for Ar diffusion in clay minerals.

- Reviewer 2:

Nothing to change. He was satisfied with the revision made before resubmitting the manuscript.

- Reviewer 3:

As per request, I have now expanded the list of references on line 67 and added two extra published articles that are indeed relevant to the brittle structural evolution of the study area. I'd like to point out, though, that my group has done pioneering work in this respect and that there is basically no available literature dealing with the reconstruction of brittle structural histories lasting several hundreds of million years. The articles added now are more limited in the scope of their results, but the reviewer is correct in her comment and they do indeed help to frame in a wider context our own research results. As for the comment for line 282 on page 12, I have now added a sentence to the discussion that specifically mentions the results of Larsen and coworkers (2003). It is indeed useful background information to our study and well worth mentioning.